# Performance of a rapid diagnostic test for the detection of *Cryptosporidium* spp. in African children admitted to hospital with diarrhea

**Gédéon Prince Manouana**[1,2,3☯], **Eva Lorenz**[4,5,6☯], **Mirabeau Mbong Ngwese**[1,2☯], **Paul Alvyn Nguema Moure**[1], **Oumou Maiga Ascofaré**[4,5,7], **Charity Wiafe Akenten**[7], **John Amuasi**[7], **Njari Rakotozandrindrainy**[8], **Raphael Rakotozandrindrainy**[8], **Joyce Mbwana**[9], **John Lusingu**[9], **Natalie Byrne**[1,2], **Sophia Melhem**[4], **Jeannot Frejus Zinsou**[1,2], **Roméo Bayodé Adegbite**[1,2], **Benedikt Hogan**[4], **Doris Winter**[4], **Jurgen May**[4], **Peter Gottfried Kremsner**[1,2,3,10], **Steffen Borrmann**[1,2], **Daniel Eibach**[4,5☯], **Ayola Akim Adegnika**[1,2,3,10☯] *

**1** Centre de Recherches Médicales de Lambaréné, **2** German Center for Infection Research (DZIF), African partner institution, CERMEL, Gabon, **3** Institut für Tropenmedizin, Universität Tübingen, Tübingen, Germany, **4** Infectious Disease Epidemiology, Bernhard Nocht Insitute for Tropical Medicine, Hamburg, Germany, **5** German Center for Infection Research (DZIF), Hamburg-Borstel-Lübeck-Riems, Germany, **6** Institute of Medical Biostatistics, Epidemiology and Informatics, University Medical Centre of the Johannes Gutenberg University Mainz, Germany, **7** Kumasi Centre for Collaborative Research in Tropical Medicine, Kumasi, Ghana, **8** University of Antananarivo, Madagascar, **9** National Institute for Medical Research (NIMR) & University of Copenhagen, Denmark, **10** German Center for Infection Research (DZIF), partner site Tübingen, Germany

☯ These authors contributed equally to this work.
\* aadegnika@cermel.org

**Data Availability Statement:** Data are available submitted as additional information S4.

## Abstract

### Background

*Cryptosporidium* is a protozoan parasite that causes mild to severe diarrhoeal disease in humans. To date, several commercial companies have developed rapid immunoassays for the detection of *Cryptosporidium* infection. However, the challenge is to identify an accurate, simple and rapid diagnostic tool for the estimation of cryptosporidiosis burden. This study aims at evaluating the accuracy of CerTest Crypto, a commercialized rapid diagnostic test (RDT) for the detection of *Cryptosporidium* antigens in the stool of children presenting with diarrhoea.

### Methods

A cross-sectional study was conducted in four study sites in Sub-Saharan Africa (Gabon, Ghana, Madagascar, and Tanzania), from May 2017 to April 2018. Stool samples were collected from children under 5 years with diarrhoea or a history of diarrhoea within the last 24 hours. All specimens were processed and analyzed using CerTest Crypto RDT against a composite diagnostic panel involving two polymerase chain reaction (PCR) tests (qPCR and RFLP-PCR,) as the gold standard.

**Funding:** The project is supported by the DFG funded grant GZ EI 1044/1-1. AAA and MGP are members of CANTAM (EDCTP-RegNet2015- 1045) and PANDORA-ID-Net (EDCTP Grant Agreement RIA2016E-1609) networks. MGP is supported by CANTAM (EDCTP-RegNet2015- 1045). The funders had no role in study design, data collection and analysis, decision to publish, or preparation of the manuscript.

**Competing interests:** The authors have declared that no competing interests exist.

## Results

A total of 596 stool samples were collected. Evaluation of the RDT yielded a very low overall sensitivity of 49.6% (confidence interval (CI) 40.1–59.0), a specificity of 92.5% (CI 89.8–94.7), positive predictive value of 61.3% (CI 50.6–71.2), and negative predictive value of 88.5% (85.3–91.1) when compared to the composite reference standard of qPCR and RFLP-PCR for the detection of *Cryptosporidium* species. Moreover, the performance of this test varied across different sites.

## Conclusion

The weak performance of the studied RDT suggests the need to carefully evaluate available commercial RDTs before their use as standard tools in clinical trials and community survey of *Cryptosporidium* infections in pediatric cohorts.

### Author summary

Diarrhoea is a common cause of death among children younger than 5 years. Treatment is based on oral rehydration and sometimes the administration of antibiotics. Several pathogens are responsible for diarrhoea in small children. *Cryptosporidium* species are one of the common pathogens causing prolonged and persistent diarrhoea, malnutrition and growth deficits among immunocompetent children and of severe diarrhoea in immunocompromised persons.

Laboratory diagnosis of cryptosporidiosis is usually achieved by microscopic detection of *Cryptosporidium* oocysts in stool specimens; staining techniques include acid-fast stains and immunofluorescence. Given that appropriate treatment is impeded by the lack of timely and accurate standard diagnostics in middle and low-income countries, rapid diagnostic tests at the point of care potentially offer a shorter time to adequate care. We evaluated a new RDT targeting *Cryptosporidium* species. This new RDT showed variable and insufficient sensitivity among children admitted to a hospital with diarrhea at four different study sites in Gabon, Ghana, Madagascar, and Tanzania.

However, this new RDT could not be used as an appropriate tool for a reliable diagnosis of Cryptosporidiosis to guide community-based screening programs.

## Introduction

Diarrheal disease accounts for one in ten cases of death among children younger than five years [1]. Cryptosporidiosis is caused by a coccidian parasite belonging to the genus *Cryptosporidium* and has been recognized as one of the major causes of diarrheal disease worldwide [2]. Several species of *Cryptosporidium* are present in several host where they can cause disease. The First reports of human illness occured in 1976, when *Cryptosporidium* was identified in rectal biopsy specimens of a 3-year-old child [3–4]. *Cryptosporidium parvum and Cryptosporidium hominis* have been reported as the major causes of persistent diarrhea in developing countries, recognized as an opportunistic disease in HIV/AIDS patients, but also responsible for large outbreaks in immunocompetent individuals in developed countries [5–6]. *Cryptosporidium spp* therefore constitute a public health concern particularly due to reports of outbreaks in day care centers, immunocompromised patients and in waterborne transmissions [7–9]. The main symptoms of Cryptosporidiosis include; watery diarrhea which may be profused or prolonged, nausea, vomiting and low-grade fever [10–11]. In developing countries, other

etiology related to the disease includes malnutrition and growth deficits among immunocompetent children and severe diarrhoea in immunocompromised persons [12].

Numerous diagnostic techniques have been used to detect *Cryptosporidium* infection in humans and animals. They include the use of faecal smears stained by the modified Ziehl–Neelsen technique, Enzyme-linked immunosorbent assays (ELISA) and polymerase chain reaction (PCR) assays [13–14]. A comprehensive overview into the detection and molecular characterization of *Cryptosporidium* has been described elsewhere [15]

The above mentioned methods have drawbacks such as being time-consuming and relatively expensive, as well as requiring well-equipped laboratories and well-trained or skilled personnel [14–16].

To overcome these limitations, antigens based-tests mostly relying on immuno-chromatographic assays have been marketed for rapid detection of *Cryptosporidium* antigens. RDTs have become increasingly popular tools and are highly suitable for point of care testing. They allow detection of antigens of one or more protozoan parasites in a single test format by lateral flow immunochromatographic assay. The advantage of such assay is that they are fast, easy to perform and interpret and thus can be used in low resource settings. Several of these RDTs have been in used over the years with varying sensitivities and specificities. In comparing the performance of 4 RDTs including; ImmunoCardSTAT! CGE, Crypto/Giardia Duo-Strip, RIDA QUICK *Cryptosporidium/Giardia/Entamoeba* Combi and *Giardia/Cryptosporidium* Quik Check the authors observed varying sensitivities ranging from 92 to 100% and 100% specificities for all RDT brands for the detection of *Cryptosporidium* when compared to ELISA, microscopy, and qPCR as gold standards [17]. In addition, these RDTs demonstrate varying performances, and some of them are not well validated for the detection of less frequent non-parvum/hominis *Cryptosporidium* species [2–18].

The CerTest Crypto is a newly commercialized RDT with manufacturer reported sensitivity and specificity of >99% respectively [19]. No studies have so far evaluated the performance of this RDT in field settings including resource-limited settings such as ours. More so, most RDTs need to be refrigerated before use, which makes them unsuitable or difficult to use in settings with varying temperatures and humidity. The CerTest Crypto RDT however does not require refrigeration.

Consequently, there is a need for an extensive evaluation of this RDT in diverse field conditions in order to evaluate their diagnostic usefulness. The aim of this study was to assess the performance of CerTest Crypto against a composite reference standard of qPCR and RFLP-PCR.

## Materials and methods

### Study design, context and site

This prospective cross-sectional study, is part of a larger cross-sectional study entitled "Genetic determinants for transmission of *Cryptosporidium parvum/hominis* among humans and animals in Africa". The study was carried out from May 2017 to April 2018 in four sub-Saharan African countries (Lambaréné in Gabon, Agogo in Ghana, Antananarivo in Madagascar and Korogwe in Tanzania). All hospitals where participants were recruited are situated in semi-urban areas across all study sites. Samples used to evaluate the test performance of CerTest Crypto were collected from these health centers.

### Study population

Stool samples were obtained from all children under 5 years of age presenting with diarrhoea or history of diarrhoea within the last 24h to the outpatient departments (OPDs) of the study hospitals of the four countries.

## Field and Laboratory procedures

Patients who provided a stool sample and an informed consent signed by the parents or legal guardian were enrolled in the study. Each child provided a fresh stool sample in a dry and leak-proof stool container. Prior to the start of the study, a general training was organized in Tanzania with participants from all the four study sites in attendance. Training was based on CerTest Crypto RDT testing and qPCR for Cryptosporidium. Laboratory technicians from all the four sites were trained on sample processing, time of testing, and protocols were developed and harmonized for use in all the four sites. The Standard Operation Pocedures (SOPs) for sample processing and testing with the CerTest Crypto RDT and qPCR were implemented in all the four sites. A small amount of stool sample (250 mg or 250 μl) was stored at -20˚C for further analyses (with qPCR, RFLP-PCR).

## CerTest Crypto RDT processing

Fresh stool samples were tested using a commercially available RDT (CerTest BIOTEC S.L, Pol. Industial Rio Gallego II, Zaragoza Spain). The RDTs were stored at room temperature in all the sites according to manufacturer instructions. The stool samples were collected and transported in cool boxes (4˚C) to the laboratory. All samples were analysed within 24 hours of sample collection. Stool sample aliquots were stored at -80˚C for further analysis. Approximately 125 mg or 125 μl of stool sample was transferred into the diluent provided by the kit manufacturer by use of an applicator stick or pipette. The suspension was homogenized by shaking. Four drops of the diluted fecal material were dispensed into the circular window of the test card. The flow was allowed to run for 10 minutes followed by a visual interpretation. The test results were not recorded later than 10 minutes as instructed by manufacturer [19].

The Certest Crypto Kit is supplied with quality control reagents. The quality control reagent consists of a known positive and negative sample. One negative and one positive control was run once a month in all the four sites and for each new batch of test kits. Additionally, the internal procedural control is included in the test, wherein the appearance of a green line in the control line result window is an indication of correct procedural technique and confirms enough specimen volume was used.

## DNA isolation

A modified MO BIO-Qiagen stool DNA extraction protocol was performed incorporating six main steps including: sample treatment, cell lysis, inhibitors removal, DNA binding, a wash step, and DNA elution. Genomic DNA was extracted from 250 mg (or 250 μL) of stool with the use of DNeasy PowerSoil Kit (QIAGEN, Hilden Germany) formerly supplied by MO BIO (MO-BIO Carlsbad, CA, USA) as PowerSoil DNA Isolation Kit. All steps of the DNA extraction were performed following the manufacturer's instructions. The nucleic acids were eluted in 100 μL volume and 5 μL of the extract was used for qPCR.

## Real-time PCR

DNA amplification was performed in a Rotor-Gene Q instrument (QIAGEN GmbH, Hilden-Germany) in 25 μL reactions using the HotStart Taq master mix kit (QIAGEN, Germany), 3.5 mM MgCl2, 500 nM forward primer (crypto-F) 5'- CGC TTC TCT AGC CTT TCA TGA -3', 500 nM reverse primer (crypto-R) 5'- CTT CAC GTG TGT TTG CCA AT-3', 175 nM Crypto probe 5'-ROX-CCA ATC ACA GAA TCA TCA GAA TCG ACT GGT ATC–BHQ2-3'. The primer and probe sequences and corresponding assession numbers have been published elsewhere [20]. These primers and probe sequences are specific for *Cryptosporidium parvum* and

*C. hominis*, although their efficiency at detecting other *Cryptosporidium* species has not be evaluated.

All samples were tested in duplicate for *Cryptosporidium* species detection employing the following cycling protocol: one cycle at 95˚C for 15 min (polymerase activation), followed by 45 cycles of 95˚C for 15 seconds (denaturation), 67˚C for 30 seconds (annealing) and 72˚C for 30 seconds (extension), followed by a final cooling step at 40˚C for 30 seconds. Phocin Herpes Virus (PhHV) Plasmid was incorporated into the master mix to control for PCR inhibitors and only samples with a cycle threshold $< = 38$ were considered positive.

### *Cryptosporidium* genotyping

All samples detected positive for *Cryptosporidium* species by qPCR were genotyped using a PCR–restriction fragment length polymorphism (RFLP) technique as described previously [21]. To improve specificity, the PCR products from the restriction digest were sequenced and resulting sequences were blasted to identify the *Cryptosporidium* species.

### Statistical analysis

Statistical analyses were performed using Stata 14. Categorical variables were described as counts and percentages. Continuous variables were described by the median and interquartile range (IQR). To ensure a comparable set of observations for the analysis, observations with missing information on either the RDT or the PCR result were excluded from the analysis and the study period was restricted from May 2017 to April 2018. We calculated sensitivity, specificity, positive predictive value (PPV) and negative predictive value (NPV) to evaluate the performance of the CerTest Crypto RDT result versus the overall results observed by combining two PCR methods (PCR, RFLP-PCR) defined as a composite reference standard PCR. The performance characteristics of CerTest Crypto RDT were determined based on the ability of the CerTest to detect all species of *Cryptosporidium* (both human and animal species). Therefore, to assess the performance characteristics of this test to detect 2 of the most frequent species(*C. hominis* and *C. parvum*), eight observations that were PCR positive for "*C. meleagridis* "and "*C. xiaoi*/*bovis*" were excluded from the analysis. PCR was positive if either method yielded a positive result, otherwise negative. The test performance measures are presented as percentages along with the respective 95% confidence intervals (CI).

### Ethical approval

Informed consent was obtained from the parents or legal guardian of the patient at the outpatient department (OPD) or inpatient department (IPD). The study protocol was approved by each Institutional Ethical Review Board of all four study sites. The National Health Research Ethics Committee (NatHREC) in Tanzania, Comité National d'Ethique du Gabon, the Committee On Human Research, Publications And Ethics, Kwame Nkrumah University Of Science And Technology of Kumasi, Ghana, the Medical Research Coordinating Committee of the National Institute for Medical Research, Tanzania (NIMR MRCC), the Ethical Committee of the Ministry of Health of the Republic of Madagascar and the Ärztekammer Hamburg, Hamburg, Germany.

## Results

A total of 596 stool samples from Ghana (N = 132), Gabon (N = 192), Madagascar (N = 83) and Tanzania (N = 189) was tested to assess the performance of CerTest Crypto RDT. Table 1 provides the frequency distribution of all *Cryptosporidium* species from all study sites

**Table 1. Frequency distribution of *Cryptosporidium* rapid diagnostic test results by study sits.**

| Country | C. hominis n (%) | C. meleagridis n (%) | C. parvum n (%) | C. xiaoi/bovis n (%) |
|---|---|---|---|---|
| **Overall (N = 482)** | 91 (18.9) | 7 (1.5) | 15 (3.1) | 1 (0.2) |
| **Ghana (N = 107)** | 15 (14) | 2 (1.9) | 7 (6.5.) | 1 (0.9) |
| **Gabon (N = 154)** | 31 (20.1) | 0 | 7 (4.5) | 0 |
| **Madagascar (N = 65)** | 14 (21.5) | 4 (6.2) | 0 | 0 |
| **Tanzania (N = 156)** | 31 (19.9) | 1 (0.6) | 1 (0.6) | 0 |

following analysis with PCR and RFLP. The diagnostic performance characteristics of *Cryptosporidium* CerTest Crypto RDT were analyzed using PCR as a reference standard in all 4 study sites. Table 2 provides sensitivity, specificity, PPV, and NPV of the test to detect *Cryptosporidium* species in the four sites. The sensitivity of the test varied considerably in all study sites being highest in Madagascar (72.22%) in comparison to Gabon and Ghana which had 50 and 52% respectively and was lowest in Tanzania with 35.29%. Conversely, the specificity of the test was similar in all study sites ranging from 86 to 94%. However, subsequent re-analyses did not result in any changes in the performance characteristic of the cerTest across study sites (Table 3).

Further analyses of all cerTest negative samples from all study sites was performed using PCR-RFLP.

To further evaluate the stability throughout the year and batches variabilities of the CerTest Crypto against PCR reference standard, we looked at the performance of the test over the course of the year in all study sites (Figs 1–4). Overall the false netative (FN) cases were detected almost every month in Gabon from the start to the end of the study period. Meanwhile, in Ghana, FN cases were detected more frequently during the first 5 months (May to September 2017) and then during the last 2 months (March to April 2018). While in Madagascar and Tanzania, the FN cases were mostly found in the beginning and towards the end of the study period. These numbers of FN observed at the four sites are not consistent and do not represent a clear pattern to suggest any batch effect on the test performance from the different study sites.

Finally, there is no evidence of the effect of gender, rainfall, sampling period as well as age group on *Cryptosporidium* infection across the study sites. However, there is a high proportion of *Cryptosporidium* infection occurring during the first two years of age accros countries (S1 Table and S1 Fig).

## Discussion

There is a need for efficient diagnostic methods for *Cryptosporidium* infection in settings where the disease is prevalent. Consequently, RDTs appear as a unique opportunity for point-of-care diagnosis in the absence of routine stool microscopy and advanced diagnostic tools. CerTest Crypto test is a commercially available RDT that is easy to perform, does not require

**Table 2. Overall and country specific test evaluation results of the RDT when compared to qPCR and RFLP as reference standard.**

| Country | Sensitivity [95% CI] | Specificity [95% CI] | Positive predictive value [95% CI] | Negative predictive value [95% CI] |
|---|---|---|---|---|
| **Overall (N = 596)** | 49.57 [40.11, 59.04] | 92.52 [89.79, 94.70] | 61.29 [50.62, 71.22] | 88.47 [85.35, 91.13] |
| **Ghana (N = 132)** | 52.00 [31.31, 72.20] | 94.39 [88.19, 97.91] | 68.42 [43.45, 87.42] | 89.38 [82.18, 94.39] |
| **Gabon (N = 192)** | 50.00 [33.38, 66.62] | 94.81 [90.02, 97.73] | 70.37 [49.82, 86.25] | 88.48 [82.60, 92.92] |
| **Madagascar (N = 83)** | 72.22 [46.52, 90.31] | 86.15 [75.34, 93.47] | 59.09 [36.35, 79.29] | 91.80 [81.90, 97.28] |
| **Tanzania (N = 189)** | 35.29 [19.75, 53.51] | 91.61 [86.08, 95.46] | 48.00 [27.80, 68.69] | 86.59 [80.40, 91.40] |

**Table 3. Overall and country specific test evaluation results of the RDT when compared to qPCR and RFLP as reference standard, excluding test results of *Cryptosporidium* meleagridis", "*Cryptosporidium* xiaoi/bovis".**

| Country | Sensitivity [95% CI] | Specificity [95% CI] | Positive predictive value [95% CI] | Negative predictive value [95% CI] |
|---|---|---|---|---|
| **Overall (N = 588)** | 49.53 [39.72, 59.37] | 92.52 [89.79, 94.70] | 59.55 [48.62, 69.83] | 89.18 [86.12, 91.77] |
| **Ghana (N = 129)** | 54.55 [32.21, 75.61] | 94.39 [88.19, 97.91] | 66.67 [40.99, 86.66] | 90.99 [84.06, 95.59] |
| **Gabon (N = 192)** | 50.00 [33.38, 66.62] | 94.81 [90.02, 97.73] | 70.37 [49.82, 86.25] | 88.48 [82.60, 92.92] |
| **Madagascar (N = 79)** | 71.43 [41.90, 91.61] | 86.15 [75.34, 93.47] | 52.63 [28.86, 75.55] | 93.33 [83.80, 98.15] |
| **Tanzania (N = 188)** | 36.36 [20.40, 54.88] | 91.61 [86.08, 95.46] | 48.00 [27.80, 68.69] | 87.12 [80.98, 91.84] |

refrigeration and saves time for the detection of *Cryptosporidium* spp. According to previous studies, the performance levels' variability of RDTs can depend on differences in commercial products, dissimilar methodologies employed and genetic diversity of *Cryptosporidium* across geographical regions [18]. The cerTest Crypto RDT, whose high-performance parameters (Sensitivity and specificity > 99%) has been claimed by the manufacturer, was evaluated in this study. Our observations on the performance characteristics of cerTest Crypto showed high specificity (Tables 1 and 2). Our findings suggest that this RDT is reliable for detecting samples with no *Cryptosporidium*. As mentioned in previous studies, this result could also mean no cross-reaction with other pathogens causing diarrhoea [18–22].

In contrast, there are different sensitivities in each study site, ranging from 35.29% to 72.22% for the detection of *Cryptosporidium* species and from 36.36% to 71.43% for the detection of *C. parvum* and *C. hominis*. This indicates that this test performs poorly in the detection of all *Cryptosporidium* species. The overall number of true positives (TP) was generally low for all four sites. Meanwhile, the number of false negative did not vary considerably by month. The trend of the distribution of *Cryptosporidium spp.* that we observed amongst the younger children is similar to that reported by Current and Garcia in 1991 [23], suggesting that the demographics could not influence the performance characteristics of cerTest Crypto (S1 Table).

Therefore, the observed low sensitivity could be due to several other factors such as low parasite densities as reported elsewhere [24–25]. Moreover, our findings show a lower apparent sensitivity of the test kit that may be as a result of comparing the test to PCR, which can detect

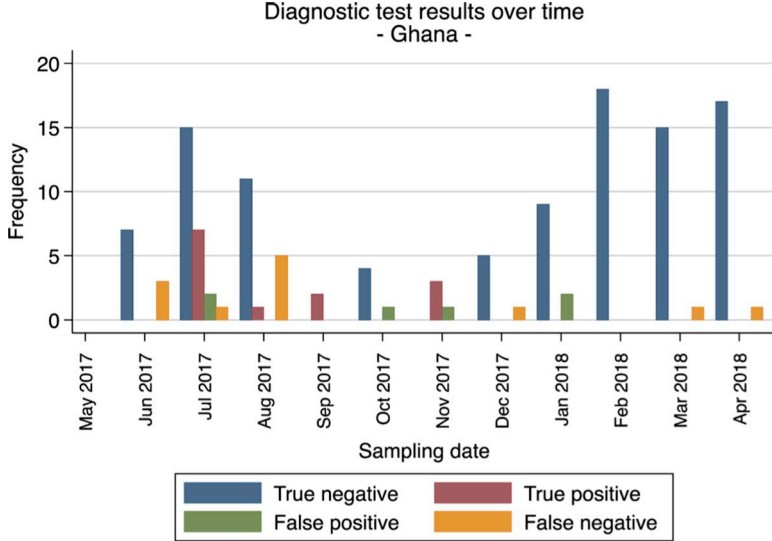

**Fig 1. Frequencies of diagnostic test evaluation (RDT vs. PCR as reference standard) from all patients for Ghana.**

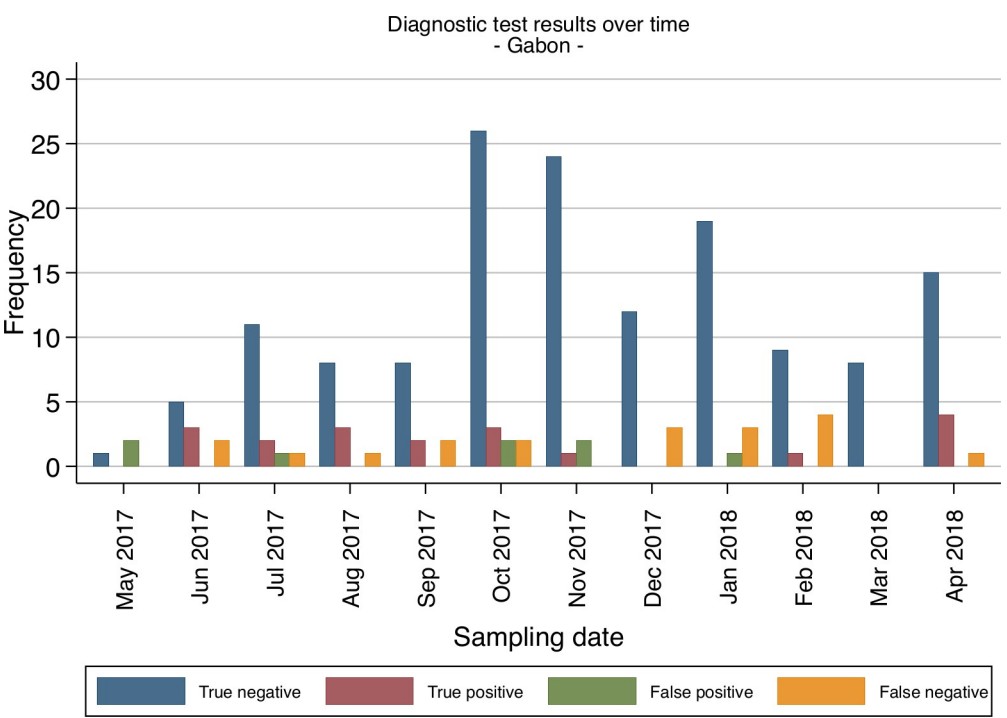

**Fig 2. Frequencies of diagnostic test evaluation (RDT vs. PCR as reference standard) from all patients for Gabon.**

parasite DNA remaining in a patient's after an infection has been cleared as previously mentioned by Boyce and O'Meara in 2017 [24]. Contrary to the study conducted by Bouyou-Akotet [25] in which the authors suggest that low sensitivity was correlated to the lower parasite load in a test that targets a specific species [26], our finding revealed a lower overall sensitivity with CerTest RDT which was not specific to one particular species.

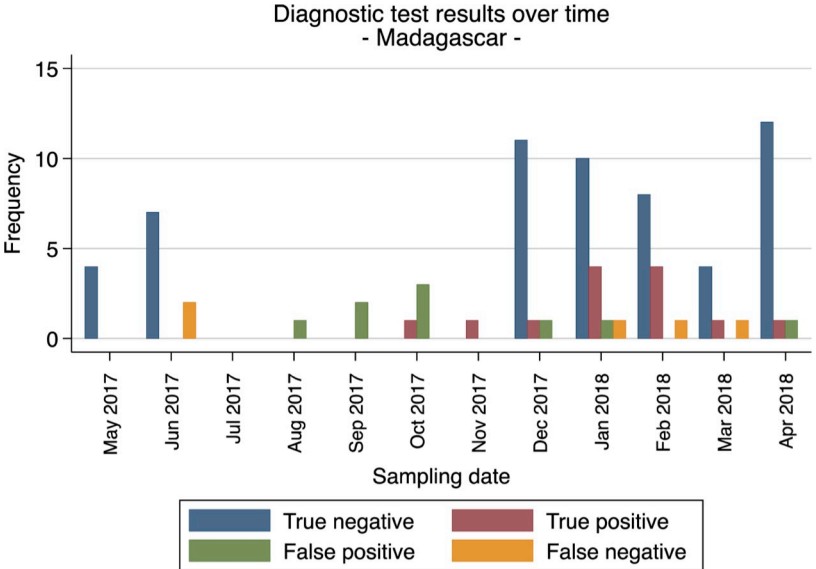

**Fig 3. Frequencies of diagnostic test evaluation (RDT vs. PCR as reference standard) from all patients for Madagascar.**

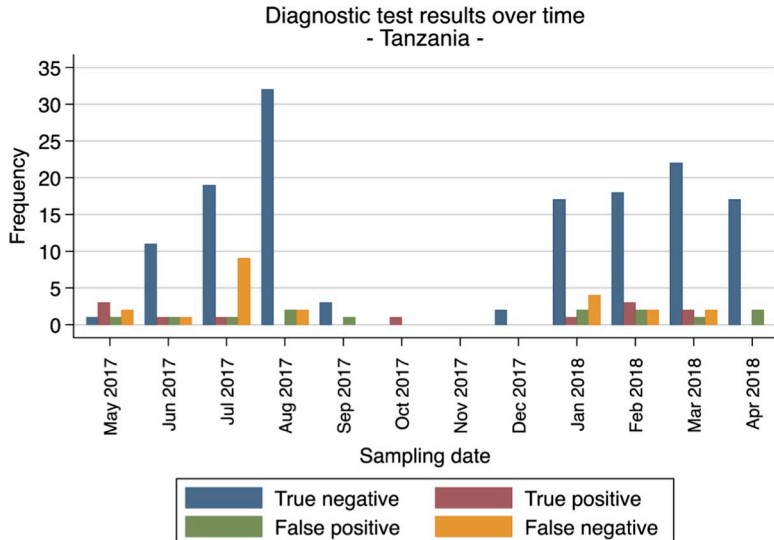

**Fig 4. Frequencies of diagnostic test evaluation (RDT vs. PCR as reference standard) from all patients for Tanzania.**

Furthermore, our finding highlighted the overall decrease in PPV due to the very low number of TP observed from the four sites. Although the NPV of 88.47% can be considered high enough, this is not comparable to manufacturer reported > 99% NPV. A considerable number of FN was observed from all study sites. Without any microscopy data to correlate the observed oocyst concentrations (infection intensities) and percentage of FN, it is difficult to determine the effect of parasite load on the detection of all species of *Cryptosporidium* (Tables 1 and 2). More so, some *Cryptosporodium* genotypes may lack part or all of the protein targeted by the CerTest RDT which could explain the observed numbers of FN. Additionally, the PPV of 61.29% is due to very low numbers of TP and a considerable number of FP observed from all sites.

The evaluation of the sensitivity of RDTs used in the detection of malaria parasites has been reported to be influenced by changing seasons [27]. In light of this, the observed differences in our case could be correlated with the fluctuation of temperatures in all study areas. Thus, most cases of FN and FP found in different months were likely due to the incidence of Cryptosporidiosis that varies with changing seasons and across geographical areas. Moreover, other reports suggest that temperatures above 30°C can affect the overall performance of the RDTs [28] in the case of malaria parasites detection. Taken together, the accuracy of any RDT result may depend on several factors such as the quality of the RDT, storage, transport and end-user performance.

Our study had two main limitations. Firstly, due to the absence of microscopy data on oocyst concentrations, we could not correlate *Cryptosporidium* oocyst load and the test performance. However, according to the manufacturer, the current test does not detect low concentration antigens in the stool. Hence a sample that tests negative with the RDT is unlikely to be positive microscopically. Taken together, we could not report the cause of all FN reported from all study sites. Secondly, with no additional investigation on protozoan pathogens causing diarrhea, we cannot exclude the possibility of cross-reacting species as the cause of all FP cases in this study. Our demographic data did not not provide any substantial evidence on the effect of temperature and humidity on the test kit. Thus the variation in performance of the test kit that we observed in this study is not related to the handling and storage of the kits.

In conclusion, CerTest Crypto RDT has been designed to detect *Cryptosporidium species*. Although this coproantigen detection assay is rapid, its sensitivity is low for the detection of *Cryptosporidium* spp as well as *C. parvum* and *C. hominis* in particular. Therefore, additional research is needed to evaluate the performance of the CerTest Crypto RDT with particular emphasis on light intensity infections whereby the concentrations of antigens present in the sample are below the detection limit of the test that may result in an increase number of false negatives.

## Supporting information

**S1 Checklist. STARD-checklist for CerTest Crypto RDT study.**
(DOCX)

**S1 Diagram. STARD diagram for participants flow in CerTest Crypto RDT study.**
(PNG)

**S1 Data. Data base_comb_reduced.**
(DTA)

**S1 Table. Proportion of *Cryptosporidium*-PCR positives and demographic data across the four study sites.**
(DOCX)

**S2 Table. Proportion of *Cryptosporidium*-PCR positives and demographic data across the four study sites.**
(DOCX)

**S1 Fig. Distribution of infected stools across sites and age in months.**
(DOCX)

## Acknowledgments

We thank all participants, laboratory staff and physicians for their professionalism and dedication.

## Author Contributions

**Conceptualization:** Oumou Maiga Ascofaré, John Amuasi, Raphael Rakotozandrindrainy, Jurgen May, Peter Gottfried Kremsner, Daniel Eibach, Ayola Akim Adegnika.

**Data curation:** Gédéon Prince Manouana, Eva Lorenz, Mirabeau Mbong Ngwese, Paul Alvyn Nguema Moure, Charity Wiafe Akenten, Njari Rakotozandrindrainy, Joyce Mbwana, John Lusingu, Natalie Byrne, Jeannot Frejus Zinsou, Roméo Bayodé Adegbite.

**Formal analysis:** Eva Lorenz, Oumou Maiga Ascofaré, Sophia Melhem, Benedikt Hogan, Doris Winter, Daniel Eibach.

**Funding acquisition:** John Amuasi, Jurgen May, Peter Gottfried Kremsner, Steffen Borrmann, Ayola Akim Adegnika.

**Investigation:** Gédéon Prince Manouana, Charity Wiafe Akenten, Doris Winter.

**Methodology:** Ayola Akim Adegnika.

**Project administration:** Gédéon Prince Manouana.

**Validation:** Mirabeau Mbong Ngwese, Ayola Akim Adegnika.

**Writing – original draft:** Gédéon Prince Manouana, Mirabeau Mbong Ngwese, Ayola Akim Adegnika.

**Writing – review & editing:** Eva Lorenz, Paul Alvyn Nguema Moure, Oumou Maiga Ascofaré, Charity Wiafe Akenten, John Amuasi, Njari Rakotozandrindrainy, Raphael Rakotozandrindrainy, Joyce Mbwana, John Lusingu, Jurgen May, Peter Gottfried Kremsner, Steffen Borrmann, Daniel Eibach.

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
