## [Decision Letter · Decision Letter 0]

13 Nov 2019

Dear Prof. Dr. Adegnika:

Thank you very much for submitting your manuscript "Performance of a rapid diagnostic test for the detection of Cryptosporidium spp. in African children admitted to hospital with diarrhea" (#PNTD-D-19-01374) for review by PLOS Neglected Tropical Diseases. Your manuscript was fully evaluated at the editorial level and by independent peer reviewers. The reviewers appreciated the attention to an important problem, but raised some substantial concerns about the manuscript as it currently stands. These issues must be addressed before we would be willing to consider a revised version of your study. We cannot, of course, promise publication at that time.

We therefore ask you to modify the manuscript according to the review recommendations before we can consider your manuscript for acceptance. Your revisions should address the specific points made by each reviewer. 

When you are ready to resubmit, please be prepared to upload the following:

(1) A letter containing a detailed list of your responses to the review comments and a description of the changes you have made in the manuscript.

(2) Two versions of the manuscript: one with either highlights or tracked changes denoting where the text has been changed (uploaded as a "Revised Article with Changes Highlighted" file); the other a clean version (uploaded as the article file).

(3) If available, a striking still image (a new image if one is available or an existing one from within your manuscript). If your manuscript is accepted for publication, this image may be featured on our website. Images should ideally be high resolution, eye-catching, single panel images; where one is available, please use 'add file' at the time of resubmission and select 'striking image' as the file type. 

Please provide a short caption, including credits, uploaded as a separate "Other" file. If your image is from someone other than yourself, please ensure that the artist has read and agreed to the terms and conditions of the Creative Commons Attribution License at http://journals.plos.org/plosntds/s/content-license (NOTE: we cannot publish copyrighted images). 

(4) If applicable, we encourage you to add a list of accession numbers/ID numbers for genes and proteins mentioned in the text (these should be listed as a paragraph at the end of the manuscript). You can supply accession numbers for any database, so long as the database is publicly accessible and stable. Examples include LocusLink and SwissProt.

(5) To enhance the reproducibility of your results, we recommend that you deposit your laboratory protocols in protocols.io, where a protocol can be assigned its own identifier (DOI) such that it can be cited independently in the future. For instructions see http://journals.plos.org/plosntds/s/submission-guidelines#loc-methods

While revising your submission, please upload your figure files to the Preflight Analysis and Conversion Engine (PACE) digital diagnostic tool, https://pacev2.apexcovantage.com/ PACE helps ensure that figures meet PLOS requirements. To use PACE, you must first register as a user. Then, login and navigate to the UPLOAD tab, where you will find detailed instructions on how to use the tool. If you encounter any issues or have any questions when using PACE, please email us at figures@plos.org.

We hope to receive your revised manuscript by Jan 12 2020 11:59PM. If you anticipate any delay in its return, we ask that you let us know the expected resubmission date by replying to this email.

To submit a revision, go to https://www.editorialmanager.com/pntd/ and log in as an Author. You will see a menu item call Submission Needing Revision. You will find your submission record there. 

Sincerely,

Luther A Bartelt

Guest Editor

Marcelo Ferreira

Deputy Editor

In response to the important methodological and results questions raised by the reviewers, please also address whether cycle-time of Cryptosporidium detection by PCR had any relationship to the rapid antigen diagnostic result. (ie. were there more false-negative RDT at higher cycle time qPCR detections).

Reviewer's Responses to Questions

**Key Review Criteria Required for Acceptance?**

**Methods**

-Are the objectives of the study clearly articulated with a clear testable hypothesis stated?

-Is the study design appropriate to address the stated objectives?

-Is the population clearly described and appropriate for the hypothesis being tested?

-Is the sample size sufficient to ensure adequate power to address the hypothesis being tested?

-Were correct statistical analysis used to support conclusions?

-Are there concerns about ethical or regulatory requirements being met?

Reviewer #1: This is an interesting study. The authors state that evaluating the accuracy of CerTest Crypto, a commercialized rapid diagnostic test (RDT) for the detection of Cryptosporidium antigens in the stool of children presenting with diarrhoea. A cross-sectional study was conducted in four study sites in Sub-Saharan Africa. All 596 specimens were processed and analyzed using CerTest Crypto RDT against a composite diagnostic panel involving two polymerase chain reaction (PCR) tests (qPCR and RFLP-PCR,) as the gold standard. GP60 sequencing was used to study potential Cryptosporidium subtype-dependent variation of the performance of the RDT.'

They didn’t mention anything regarding storage of their stool samples for CerTest Crypto. What was the timing of the test after collecting the stool samples? Did the test run on the same day of collecting stool samples. More details need to be given about this CerTest. From where they procured this test.

The primer-probe set used in this study was specific for C.Parvum and not for Cryptosporidium species,

Reviewer #2: Authors need to review some aspects in materials and methods as indicated below:

Ethical approval: Some important details are needed about the lab tech training both on CerTest Crypto RDT using and PCR test (one training including the lab tech of all sites? Or one training per site? The training was followed by trainer’s assessment or not?). 

No mention of quality control and quality assurance in the materials and methods part. Why? Kindly clarify.

Lines 114-116: “All hospitals where participants were recruited are situated in semi-urban areas across all study sites. All samples collected within this study were used to evaluate the test performance of CerTest Crypto compared to the composite reference standard of qPCR and RFLP-PCR”. Please, give provide clarifications (or rephrase) since this is not observed in the results (cf Tables 1 and 2).

Line 122: As indicated in Ethical approval it is better to write: The parents or legal guardian of the patient who provided the signed informed consent …..

Line 180: Please add: of Ghana (the Committee On Human Research, Publications And Ethics, Kwame Nkrumah University Of Science And Technology of Ghana, …)

Reviewer #3: See "Summary and General Comments Section" for detailed comments

-Are the objectives of the study clearly articulated with a clear testable hypothesis stated? YES

-Is the study design appropriate to address the stated objectives? YES

-Is the population clearly described and appropriate for the hypothesis being tested? NO (description inadequate)

-Is the sample size sufficient to ensure adequate power to address the hypothesis being tested? YES

-Were correct statistical analysis used to support conclusions? YES

-Are there concerns about ethical or regulatory requirements being met? YES

**Results**

-Does the analysis presented match the analysis plan?

-Are the results clearly and completely presented?

-Are the figures (Tables, Images) of sufficient quality for clarity?

Reviewer #1: The result is poorly written, the quality of experimental data need to be improved . Frequency distribution of RDT test results in Table 1. Presentation of Table 2 and Table 3 are not very clear. The authors are requested to compare . Sensitivity, Specificity, Positive Predictive Value (PPV) and, Negative Predictive Value (NPV) of RDT when compared with qPCR assay should be mentioned.

Reviewer #2: The analysis presented match the analysis plan. However many complementary informations will need to clarify some points. 

Lines 185-188: Please, remove this sentence here and completed materials and methods with: To ensure a comparable ….. to April 2018. 

Line 192: Please, replace all 4 sites by the four sites.

Lines 196-200: Please remove “These performance characteristics …were excluded from the analysis”. Better in materials and methods part.

Discussion

Your study shows intra sites sensitivity compared to Rapid test specificity (The sensitivity of the test varied considerably in all study sites being highest in Madagascar (72.22%) in comparison to Gabon and Ghana which had 50 and 52 % respectively and was lowest in Tanzania with 35.29 %). Kindly, give some eventual raisons (training? cerTest Crypto RDT conservation? cerTest Crypto RDT manufacturing? samples conservation????).

Limitations

The absence of microscopic data is a real limitations of these results

The absence of QC and QA will need to be discussed if it’s the case.

References

Please, updated the references. For example, the references below should be helpful

Bouzid et al, Clin Microbiol Rev. 2013: 115–134. doi: 10.1128/CMR.00076-12

Garcia et al, Clin Microbiol Rev. 2018: e00025-17. doi: 10.1128/CMR.00025-17

Cunha et al, Rev Inst Med Trop Sao Paulo. 2019; 61: e28. doi: 10.1590/S1678 9946201961028

Sylvia Afriyie Squire, Una Ryan, Parasit Vectors. 2017; 10: 195. doi: 10.1186/s13071-017-2111-y

Moore et al, PLoS Negl Trop Dis. 2016 Jul; 10(7): e0004822. Published online 2016 Jul 7. doi: 10.1371/journal.pntd.0004822

……………………..

Reviewer #3: See "Summary and General Comments Section" for detailed comments

-Does the analysis presented match the analysis plan? YES

-Are the results clearly and completely presented? YES

-Are the figures (Tables, Images) of sufficient quality for clarity? YES

**Conclusions**

-Are the conclusions supported by the data presented?

-Are the limitations of analysis clearly described?

-Do the authors discuss how these data can be helpful to advance our understanding of the topic under study?

-Is public health relevance addressed?

Reviewer #1: Conclusion is not very clear by the data presented.

Reviewer #2: The conclusions of this study supported the data presented but the limitations of analysis didn't clearly described.

The authors discuss how these data can be helpful to advance our understanding of the topic under study and the public health relevance was addressed

Reviewer #3: See "Summary and General Comments Section" for detailed comments

-Are the conclusions supported by the data presented? YES

-Are the limitations of analysis clearly described? PARTIALLY

-Do the authors discuss how these data can be helpful to advance our understanding of the topic under study? YES

-Is public health relevance addressed? YES

**Editorial and Data Presentation Modifications?**

Reviewer #1: Many changes will be required

Reviewer #2: No comment

Reviewer #3: See "Summary and General Comments Section" for detailed comments

**Summary and General Comments**

Reviewer #1: This is paper of interest but, poorly written and need to be revised

Reviewer #2: Most human infections causing diarrhea are attributed to two species: Crytosporidium parvum (zoonotic transmission) and C. hominis (anthroponotic transmission). Overall this is an important paper for improving epidemiology and diagnostic of Cryptosporidium parasites using CerTest Crypto rapid diagnostic test; therefore could lead to better and faster management of diarrhea cases in our hospitals. 

I have a few general comments. Please add in the introduction the background of the different cryptosporidium species in the four countries and clearly update diagnostic tests validated by the WHO to date. Although well readable, the text would benefit from a review by a professional editors include as co-authorships. The English is not fluent in several passages and it looks like the text has been translated from French.

Reviewer #3: Summary statement: The authors presents an analysis of the efficacy of a novel Cryptosporidium rapid diagnostic test. The manuscript is clear and addresses an important concern – specifically, that RDTs may be efficacious in the lab but their reliability and utility in the field is more important. The authors qualitatively evaluate the potential ‘batch effect’ variability in the assay; however, relevant details about the study and study design should be included. There may be a demographic bias rather than batch bias of the test. For example, the assay may be more efficacious when testing in young infant (i.e. using age as a proxy for parasitemia), vs. children and adults. Alternatively, it might be more effective in children who are at low risk for multiple infections rather than high risk for multiple infections (i.e. addressing the author’s concern about off target organisms). By presenting more of the study design and cohort demographic data, these considerations could be evaluated.

Major concerns:

1) Is this the first report of this cross-sectional study (or the cohort represented in the study)? If so, greater methodological detail is necessary. If not, please include a citation.

2) Please include an analysis of patient demographics, per my point in the summary statement.

3) Please clarify in the ‘Real-time PCR’ section which species could be detected with these pan-Crypto primers.

4) Table 1 – what do the dashes mean? If 0, please replace with 0.

5) Is the protein targeted in the CerTest assay known? Can it be BLAST-ed to identify which organisms might be responsible for the hypothesized cross-reaction from other protozoa?

6) On lines 96-98, the authors mention that RDTs tend to have reduced sensitivity with uncommon species. What species are defined as ‘uncommon’? Naming the species would make it clear if this problem motivated Table 2.

7) Line 308 – without knowing more about the RDT, the following statement is not clear: ‘observed color intensity on the test window in case of light infections’. It sounds as if the authors are suggesting that the intensity of the test color band could be difficult to see in light infections. The manufacturer’s instructions available online do not include any text related to interpreting presence/absence of the band in cases where intensity is questionable. The authors should update the methods section to include details of the training provided to field workers related to interpretation of the CerTest results.

Minor concerns: 

1) Please add references for claims about the assay (Around line 100)

2) Line 133 – Please provide a reference for the manufacturer’s instructions.

3) Line 135 – ‘modified MO BIO-Qiagen stool DNA extraction protocol’ Were both the MO-BIO and Qiagen versions of the kit used, and if so could there be an extraction bias with the different versions?

Grammar/copy edit suggestions: 

Lines 71-73: ‘Cryptosporidiosis which is caused by a coccidian parasite belonging to the genus Cryptosporidium has been recognized as one of the major causes of diarrheal disease worldwide (2).’ – this is grammatically incorrect. I’d suggest replacing it with ‘Cryptosporidiosis is caused by a coccidian parasite belonging to the genus Cryptosporidium and has been recognized as one of the major causes of diarrheal disease worldwide (2).’

Lines 82-84: ‘All of these methods have some drawbacks that include; being time-consuming and relatively expensive, require well-equipped laboratories and well-trained or skilled personnel’ – Inappropriate use of semi colon. I’d suggest replacing this text with ‘All of these methods have drawbacks including being time-consuming and relatively expensive, as well as requiring well-equipped laboratories and well-trained or skilled personnel’

line 98 – Please add new paragraph character.

Line 125: define SOP.

Line 136 – ‘Inhibitors’ should be ‘inhibitors’.

A number of abbreviations are presented that are unnecessary (i.e. not ever used again).

Cryptosporidium is periodically not italicized.

Line 304 – ‘species’ should be ‘species’.

PLOS authors have the option to publish the peer review history of their article (what does this mean?). If published, this will include your full peer review and any attached files.

Reviewer #1: No

Reviewer #2: No

Reviewer #3: No

---

## [Decision Letter · Decision Letter 1]

20 Feb 2020

Dear Prof. Dr. Adegnika,

Thank you very much for submitting your manuscript "Performance of a rapid diagnostic test for the detection of Cryptosporidium spp. in African children admitted to hospital with diarrhea" for consideration at PLOS Neglected Tropical Diseases. As with all papers reviewed by the journal, your manuscript was reviewed by members of the editorial board and by several independent reviewers. The reviewers appreciated the attention to an important topic. Based on the reviews, we are likely to accept this manuscript for publication, providing that you modify the manuscript according to the review recommendations. 

Sincerely,

Luther A Bartelt

Guest Editor

Marcelo Ferreira

Deputy Editor

Reviewer's Responses to Questions

**Key Review Criteria Required for Acceptance?**

**Methods**

-Are the objectives of the study clearly articulated with a clear testable hypothesis stated?

-Is the study design appropriate to address the stated objectives?

-Is the population clearly described and appropriate for the hypothesis being tested?

-Is the sample size sufficient to ensure adequate power to address the hypothesis being tested?

-Were correct statistical analysis used to support conclusions?

-Are there concerns about ethical or regulatory requirements being met?

Reviewer #2: Authors described clearly the study methods with objectives clearly articulated. Study population well described with adequate sample size power. Endeed, the conclusions were supported by correct statistical analysis.

Reviewer #3: Yes

**Results**

-Does the analysis presented match the analysis plan?

-Are the results clearly and completely presented?

-Are the figures (Tables, Images) of sufficient quality for clarity?

Reviewer #2: In this study all analysis match the analysis plan and the results are presented accordingly with good tables and figures.

Reviewer #3: Yes

**Conclusions**

-Are the conclusions supported by the data presented?

-Are the limitations of analysis clearly described?

-Do the authors discuss how these data can be helpful to advance our understanding of the topic under study?

-Is public health relevance addressed?

Reviewer #2: In this reviewed version, the conclusions are supported by data presented and authors discuss how data can be helpful to advance the study topic understanding. However some limitations will need to be clarify (ex. microscopic data)

Reviewer #3: Yes

**Editorial and Data Presentation Modifications?**

Reviewer #2: This manuscript can be accepted for publication in PloS NTD.

Reviewer #3: (No Response)

**Summary and General Comments**

Reviewer #2: No comment

Reviewer #3: We find the scientific content of the manuscript acceptable in its current state, but clarify that our previous concern about demographics bias was raised because there is substantial variation in the incidence of crypto infection even within the first five years of life. Cryptosporidium detection, and cryptosporidium-positive stools, are heavily biased towards the first year of life and incidence is well known to be higher earlier in life (see Cryptosporidiosis, Current & Garcia, Clinical Microbiology Reviews 1991). There is no scientific value of discussion of batch effects if it does not also discuss potential demographic biases (with known associations with crypto incidence) across the batches. We recommend including demographic summaries across sites for factors known to be associated with incidence of crypto (gender balance, average age at sampling, average monthly rainfall at each site).

PLOS authors have the option to publish the peer review history of their article (what does this mean?). If published, this will include your full peer review and any attached files.

Reviewer #2: No

Reviewer #3: No
---

## [Decision Letter · Decision Letter 2]

23 Mar 2020

Dear Prof. Dr. Adegnika,

Thank you very much for submitting your manuscript "Performance of a rapid diagnostic test for the detection of Cryptosporidium spp. in African children admitted to hospital with diarrhea" for consideration at PLOS Neglected Tropical Diseases. As with all papers reviewed by the journal, your manuscript was reviewed by members of the editorial board and by several independent reviewers. The reviewers appreciated the attention to an important topic. Based on the reviews, we are likely to accept this manuscript for publication, providing that you modify the manuscript according to the review recommendations. 

Sincerely,

Luther A Bartelt

Guest Editor

Marcelo Ferreira

Deputy Editor

Reviewer's Responses to Questions

**Key Review Criteria Required for Acceptance?**

**Methods**

-Are the objectives of the study clearly articulated with a clear testable hypothesis stated?

-Is the study design appropriate to address the stated objectives?

-Is the population clearly described and appropriate for the hypothesis being tested?

-Is the sample size sufficient to ensure adequate power to address the hypothesis being tested?

-Were correct statistical analysis used to support conclusions?

-Are there concerns about ethical or regulatory requirements being met?

Reviewer #3: Demographic information as reported in the new Supplementary Table S4 are inconsistent. Based on sex only being indicated for ~10% of each population and age for less than ~20% for each population, it appears the authors have only indicated demographic information for the crypto-positive children. Demographic information from all tested children is necessary to determine whether differences in age or sex across cohorts influenced test performance. The authors should indicate the total number of male and female subjects within each site, note just the number of males and females that were PCR-positive. A similar breakdown should be included for age. Additionally, a simple histogram with ~2 month resolution would easily demonstrate whether there are substantial differences in the distribution of subject age in each cohort.

**Results**

-Does the analysis presented match the analysis plan?

-Are the results clearly and completely presented?

-Are the figures (Tables, Images) of sufficient quality for clarity?

Reviewer #3: Yes

**Conclusions**

-Are the conclusions supported by the data presented?

-Are the limitations of analysis clearly described?

-Do the authors discuss how these data can be helpful to advance our understanding of the topic under study?

-Is public health relevance addressed?

Reviewer #3: Yes, except for the statements related to Age and Sex differences across cohorts.

**Editorial and Data Presentation Modifications?**

Reviewer #3: (No Response)

**Summary and General Comments**

Reviewer #3: (No Response)

PLOS authors have the option to publish the peer review history of their article (what does this mean?). If published, this will include your full peer review and any attached files.

Reviewer #3: No
---

## [Editor Report · Decision Letter 3]

7 May 2020

Dear Prof. Dr. Adegnika,

Thank you very much for submitting your manuscript "Performance of a rapid diagnostic test for the detection of Cryptosporidium spp. in African children admitted to hospital with diarrhea" for consideration at PLOS Neglected Tropical Diseases. As with all papers reviewed by the journal, your manuscript was reviewed by members of the editorial board and by several independent reviewers. The reviewers appreciated the attention to an important topic. Based on the reviews, we are likely to accept this manuscript for publication, providing that you modify the manuscript according to the review recommendations. 

Thank you for your most recent revision. Please update the rebuttal letter and address questions in the supplementary material as requested by the reviewer. 

I apologize for the time delays in our response as Covid-19 operations in our hospital have been very time-consuming.

Sincerely,

Luther A Bartelt

Guest Editor

Marcelo Ferreira

Deputy Editor

Thank you for your most recent revision. Please update the rebuttal letter and address questions in the supplementary material as requested by the reviewer. 

I apologize for the time delays in our response as Covid-19 operations in our hospital have been very time-consuming.
---

## [Decision Letter · Decision Letter 4]

2 Jun 2020

Dear Prof. Dr. Adegnika,

We are pleased to inform you that your manuscript 'Performance of a rapid diagnostic test for the detection of Cryptosporidium spp. in African children admitted to hospital with diarrhea' has been provisionally accepted for publication in PLOS Neglected Tropical Diseases.

Best regards,

Luther A Bartelt

Guest Editor

Marcelo Ferreira

Deputy Editor

Reviewer's Responses to Questions

**Key Review Criteria Required for Acceptance?**

**Methods**

-Are the objectives of the study clearly articulated with a clear testable hypothesis stated?

-Is the study design appropriate to address the stated objectives?

-Is the population clearly described and appropriate for the hypothesis being tested?

-Is the sample size sufficient to ensure adequate power to address the hypothesis being tested?

-Were correct statistical analysis used to support conclusions?

-Are there concerns about ethical or regulatory requirements being met?

Reviewer #3: (No Response)

**Results**

-Does the analysis presented match the analysis plan?

-Are the results clearly and completely presented?

-Are the figures (Tables, Images) of sufficient quality for clarity?

Reviewer #3: (No Response)

**Conclusions**

-Are the conclusions supported by the data presented?

-Are the limitations of analysis clearly described?

-Do the authors discuss how these data can be helpful to advance our understanding of the topic under study?

-Is public health relevance addressed?

Reviewer #3: (No Response)

**Editorial and Data Presentation Modifications?**

Reviewer #3: (No Response)

**Summary and General Comments**

Reviewer #3: The authors have addressed all of our concerns

PLOS authors have the option to publish the peer review history of their article (what does this mean?). If published, this will include your full peer review and any attached files.

Reviewer #3: No

---

## [Editor Report · Acceptance letter]

30 Jun 2020

Dear Prof. Dr. Adegnika,

We are delighted to inform you that your manuscript, "Performance of a rapid diagnostic test for the detection of Cryptosporidium spp. in African children admitted to hospital with diarrhea," has been formally accepted for publication in PLOS Neglected Tropical Diseases.

Best regards,

Shaden Kamhawi

co-Editor-in-Chief

Paul Brindley

co-Editor-in-Chief
